# Ventricular and Atrial Remodeling after Transcatheter Edge-to-Edge Repair: A Pilot Study

**DOI:** 10.3390/jpm12111916

**Published:** 2022-11-16

**Authors:** Alessandro Albini, Matteo Passiatore, Jacopo Francesco Imberti, Anna Chiara Valenti, Giulio Leo, Marco Vitolo, Francesca Coppi, Fabio Alfredo Sgura, Giuseppe Boriani

**Affiliations:** 1Cardiology Division, Department of Biomedical, Metabolic and Neural Sciences, University of Modena and Reggio Emilia, Policlinico di Modena, 41124 Modena, Italy; 2Clinical and Experimental Medicine PhD Program, University of Modena and Reggio Emilia, 41121 Modena, Italy

**Keywords:** mitral regurgitation, percutaneous mitral valve repair, atrial fibrillation, echocardiography, reverse remodeling

## Abstract

Background: The aim of this study was to determine the impact of transcatheter edge-to-edge repair (TEER) on left and right ventricular (LV, RV) and left and right atrial (LA, RA) remodeling according to the mechanism of mitral regurgitation (MR) and history of atrial fibrillation (AF). Methods: Twenty-four patients (mean age 78.54 years ± 7.64 SD; 62.5% males) underwent TEER at our center. All the patients underwent echocardiography 1.6 ± 0.9 months before the procedure and after 5.7 ± 3.5 months; functional MR accounted for 54% of cases. Results: Compared to baseline, a statistically significant improvement in LV end-diastolic diameter (LVEDD), LV indexed mass (ILVM), LV end-diastolic and end-systolic volumes (LVEDV, LVESV), indexed LA volume (iLAV), and morpho-functional RV parameters was recorded. LVEDD and LVEDV improved in primary MR cohort, whereas in secondary MR, a significant reduction in LVEDV and LVESV was found without a significant functional improvement. LA reverse remodeling was found in organic MR with a trend toward ameliorated function. Furthermore, a significant reduction of LA volumetry was detected only in patients without history of AF (AF baseline 51.4 mL/m^2^ IQR 45.6–62.5 mL/m^2^ f-u 48.9 mL/m^2^ IQR 42.9–59.2 mL/m^2^; *p* = 0.101; no AF baseline 43.5 mL/m^2^ IQR 34.2–60.5 mL/m^2^ f-u 42.0 mL/m^2^ IQR 32.0–46.2 mL/m^2^; *p* = 0.012). As regards right sections, the most relevant reverse remodeling was obtained in patients with functional MR with a baseline poorer RV function and more severe RA and RV dilation. Conclusion: TEER induces reverse remodeling involving both left and right chambers at mid-term follow-up. To deliver a tailored intervention, MR mechanism and history of AF should be considered in view of the impact on remodeling process.

## 1. Introduction

Mitral regurgitation (MR) is the second-most common valvular disease in Europe. Chronic severe MR leads to the remodeling of left and right heart with left ventricular (LV) and atrial (LA) dilatation and dysfunction and the development of a significant pulmonary hypertension in almost half of the patients [1]. The increase in right ventricular (RV) afterload induces the remodeling of right chambers with subsequent RV dysfunction and a negative impact on patient prognosis [2].

Mitral valve repair or replacement represents the standard of care for patients with significant MR and, when performed in time, promotes LV and RV reverse remodeling improving the prognosis of patients [3].

Unfortunately, a significant percentage of patients with severe MR are not suitable for surgical mitral valve repair or replacement due to prohibitive surgical risk and/or comorbidities. In this setting, transcatheter edge-to-edge repair (TEER) has been shown to be an effective and safe alternative associated with a favorable clinical outcome [4]. Although some data on the reverse remodeling and function of heart chambers following TEER are available [5,6,7,8], we think that further research and more reliable data should be collected to reach a more detailed comprehension of the left and right section interplay in the process. Advanced echocardiography with the implementation of deformational speckle-tracking imaging (STI) could represent the best tool for the evaluation of left and right cardiac chamber remodeling by combining morphological and functional evaluations. In the present observational study, we aimed to evaluate left and right chamber reverse remodeling in a cohort of patients with severe organic and functional MR after TEER with the MitraClipTM system, evaluated with standard 2D echocardiography and STI.

## 2. Materials and Methods

In this monocentric study, we retrospectively reviewed 35 patients with severe organic or functional MR referred for TEER at Modena University Hospital, between November 2017 and September 2021. Patients with inadequate echocardiographic views for STI were excluded (11 patients), for a final cohort of 24 patients. All TEER were performed with the MitraClipTM system. Appropriateness for TEER was evaluated by a local heart team as indicated by current guidelines [9], and no patients were considered eligible for surgical valve replacement by the heart team. The suitability for MitraClipTM treatment was assessed according to the EVEREST eligibility criteria [10]. For functional MR, LV dilation and function and the severity of MR were evaluated according to inclusion criteria reported in COAPT trial [11]. Anthropometric parameters, clinical history, and medical therapy were collected for all the patients. Patients were on stable, optimized, individual-targeted heart failure therapy and underwent percutaneous coronary angioplasty when appropriate. The study was conducted according to the guidelines of the Declaration of Helsinki and approved by the Ethics Committee of Azienda Ospedaliero-Universitaria di Modena.

### 2.1. Echocardiographic Assessment

All patients underwent baseline 2D and STI echocardiographic evaluation, which is the same evaluation used to assess cardiac remodeling was performed 6 months after the procedure. All echocardiographic studies were performed with dedicated equipment and ECG gating. The left lateral decubitus images were acquired at a variable depth of 14–20 cm. Data were acquired on the parasternal long axis, short axis, and apical projections (2, 3, 4, 5 chambers). Indexing was carried out for the body surface area (BSA) according to Mosteller equation. LV end-diastolic volume (LVEDV), LV end-systolic volume (LVESV), LV ejection fraction (LVEF), and LA volumes were measured in the 4- and 2-chamber views using the biplane Simpson’s method, as recommended by international guidelines [12]. The longitudinal deformation parameters of the free wall of the RV (FWRV) and LA were also analyzed. STI analyses were performed using commercially available AutoStrain LA and AutoStrain RV softwares (Philips Healthcare). The longitudinal strain of the RV free wall (RV-FWLS) was obtained from the standard bidimensional grayscale image of an RV-focused apical 4-chamber view (framerate 50–70 Hz). There were six regions of interest (ROI) in the 4-chambers view to assess the RV global longitudinal strain. However, aiming at better reproducibility of the RV-FWLS, we chose to restrict the ROI to the basal, mid-cavity, and apical segments. For LA evaluation, bidimensional grayscale images were acquired in a 4-chamber apical projection (framerate 50–110 Hz). The auricle and pulmonary veins were not included. The peak left atrial longitudinal strain (PALS) was automatically assessed by the software [13]. Due to the significant proportion of eccentric MR in the population, a semi-quantitative method with color Doppler was employed for the baseline evaluation and the post-implant grading.

### 2.2. Statistical Analysis

Continuous variables were expressed as mean ± standard deviation (SD) for normal distributions; median and interquartile ranges (IQR) were used for non-normal distributions. Categorical data were reported as numbers and percentages. The normality of the parameters tested was evaluated using the Shapiro–Wilk test. Due to the small size of the sample, continuous variables were compared using the Wilcoxon matched pairs signed ranks test. A two-side *p* value of < 0.05 was considered significant for all analyses. Statistical analyses were performed using IBM SPSS Statistics (IBM Corp. Released 2020. IBM SPSS Statistics for Windows, Version 27.0. Armonk, NY, USA: IBM Corp.).

## 3. Results

### 3.1. Patient Population

A total of 24 high-surgical risk patients undergoing TEER for severe MR were included (mean age 78.54 years ± 7.64 SD; 15 (62.5%) were males). Baseline echocardiographic evaluation was performed 1.6 ± 0.9 months before the procedure; all patients underwent follow-up echocardiography 5.7 ± 3.5 months after the procedure. The median STS score value was 3.5% [IQR 1.7–4.8%] and the median Euroscore II was 4% [IQR 3.1–6.2%]. Organic and functional etiologies of MR were equally represented in the population at 45.8% vs. 54.2%, respectively. All patients were severely symptomatic (New York heart Association (NYHA) class III 17/24, 70.8%; NYHA class IV 7/24, 29.2%), despite optimized guideline-directed heart failure medical therapy. Baseline demographic and clinical characteristics are detailed in Table 1. As reported in the tables, no significant differences in clinical characteristics were observed in the functional and organic sub-cohorts, except for sex distribution and mean BSA. Patients with a history of AF had persistent long-term or permanent AF, so baseline and follow-up echocardiographic assessment were performed during AF.

### 3.2. Clinical and Valve Regurgitation Assessment

After TEER, there was a statically significant improvement in the NYHA class (median pre-NYHA class III vs median post-NYHA class II; *p* < 0.001). A positive procedural result in terms of reduction of the degree of MR was reached in all the subjects (Figure 1). No significant changes in the medical therapy were reported before and after the procedure. (Appendix A).

### 3.3. Baseline Echocardiographic Characteristics

Baseline echocardiographic characteristics are detailed in Figure 2 and Table 2. Indexed LVED volume was found to be dilated in 63% of patients (median iLVEDV-2D 67.5 mL/m^2^). LA dilation was found in almost all patients (median iLAV-2D 50.5 mL/m^2^). Elevated tricuspid valve regurgitation maximal velocity (TR Vmax 2.9 (2.5–3.1) m/s) and increased estimated peak systolic pulmonary artery pressure (EPSPAP 40.0 (35.0–50.0) mmHg) were found in most cases. In the baseline comparison between functional and organic MR subgroups LVESV, LVEF, and TAPSE were found to be significantly more compromised in the functional cohort. 

### 3.4. LV Size and Function during Follow-Up

In the overall cohort, after a mean follow-up period of 5.7 ± 3.5 months, a statistically significant reduction was observed in left ventricular end-diastolic diameter (LVEDD) (baseline LVEDD 57.5 mm vs. post-procedural LVEDD 55.0 mm) but not in LVES diameter (LVESD). Similarly, indexed LVED and LVES volumes significantly improved (baseline 2D-iLVEDV 67.5 mL/m^2^ vs. post-procedural 2D-iLVEDV 53.5 mL/m^2^ and baseline 2D-iLVESV-2D 35.4 mL/m^2^ vs. post-procedural 2D-iLVESV 27.1 mL/m^2^, respectively). On the contrary, none of the parameters of LV systolic function showed a significant improvement during the follow-up period (baseline 2D-LVEF 48.8% vs. post-procedural 2D-LVEF 49.6%; baseline systolic velocity at the mitral annular (LV S’ TDI) 8.0 cm/s vs. post-procedural LV S’ TDI 8.0 cm/s) (Table 3). When considering the organic MR cohort, a reduction in LVEDD was still observed with consequently reduction in estimated LV indexed mass and a significant reduction in LVEDV was still reported. In this subset of patients with baseline preserved LVEF and normal LVESV, no significant improvement in terms of functional parameters was reported. When considering the functional subgroup, no significant reduction in LVEDD was reported; conversely, a significant reduction in LVEDV and LVESV was reported with a small although not significant improvement of the LVEF (Figure 3 and Appendix A).

### 3.5. LA Size and Function during Follow-Up

A statistically significant reduction in LA maximum volume was observed after TEER procedure (baseline 2D-iLAV 50.5 mL/m^2^ vs. post-procedural 2D-iLAV 44.1 mL/m^2^). Atrial function assessed by PALS improved during the follow-up period but without reaching statistical significance (baseline PALS 11% vs. post-procedural PALS 14.6%), see Table 3. These changes were reported in the organic subset of patients with mild LA dilation and LA reservoir function reduction at baseline. In the functional cohort, with more dilated atria and with a severely reduced reservoir function at the basal evaluation, no statistically significant reduction in LA volumes and a worsening LA reservoir function were observed at follow-up (Figure 3). When comparing patients with and without history of atrial fibrillation (AF), a significant reduction of LA volumetry was detected only in patients without anamnestic AF (Figure 4 and Appendix A).

### 3.6. RV Size and Function during Follow-Up

A significant reduction in RV diameters (RVD)-basal, mid-cavity, and longitudinal-was observed with 2D echocardiography (baseline basal-RVD 38.5 mm vs. post-procedural basal-RVD 33.5 mm; baseline mid-cavity RVD 29.5 mm vs. post-procedural mid-cavity RVD 27 mm; baseline longitudinal RVD 64 mm vs. post-procedural longitudinal RVD 58.5 mm). Moreover, a significant improvement in RV systolic function’s parameters was noted in terms of fractional area change (FAC) (baseline FAC 35.5% vs. post-procedural FAC 44%), tricuspid annular plane systolic excursion (baseline TAPSE 14 mm; post-procedural TAPSE 18.5 mm), and RV free wall longitudinal strain (baseline RV-FWLS -19%; post-procedural RV-FWLS -21%).

Regarding RV after-load dependent parameters, a significant reduction in TR jet max velocity and derived PAPs was recorded during the follow-up period (baseline TR Vmax 2.9 m/s; post-procedural 2.2 m/s and baseline PAPs 40 mmHg; post-procedural 20 mmHg) (Table 3). The subanalysis of organic and functional MR showed that the most relevant reverse remodeling was obtained in patients with functional MR with a baseline poorer right ventricle function and more severe right atrial and ventricular remodeling (Figure 3 and Appendix A).

## 4. Discussion

In this pilot study of 24 patients that underwent TEER and were evaluated at baseline and after 5.69 ± 3.50 months with morpho-functional echocardiographic assessment, our main findings were as follows: (i) a significant improvement in all the RV morphological and functional parameters; (ii) a significant reduction in LVED diameters and volumes without significant improvement in LV function; (iii) significant decrease in LA volume and a trend toward an ameliorated LA function; (iv) a difference in the relative impact of TEER on cardiac remodeling when considering primary and secondary MR and subjects with and without history of AF. 

Our results add to the current body of evidence by giving a freeze-frame of short-term cardiac remodeling after TEER in a cohort of patients with equally distributed primary and secondary MR that was treated by following current guidelines and recommendations [9].

### 4.1. Right Chambers Remodeling

The most relevant result of our study is the evidence of reverse remodeling in the right sections at 6 months after TEER, with a more impactful effect in the functional MR cohort. The importance of this finding of pivotal importance due to the renowned negative prognostic impact of right involvement in valvular heart diseases and heart failure [14,15]. This is even more important when we consider the inconsistent evidence about RV dysfunction in patients undergoing surgical mitral valve repair or replacement [2,16,17,18,19,20]. TEER does not imply cardiopulmonary bypass with its possible direct negative impact on right ventricle performance and does not require pericardiotomy, making angle-dependent RV longitudinal performance parameters reliable in the postoperative setting. Evidence about early RV recovery after TEER is growing, and our results are in line with the existing literature. Giannini et al. in 2014 reported for the first time an integrated evaluation of left and RV function after TEER in functional MR [21]. RV functional parameters and PAPs resulted in amelioration at 6 months after the procedure along with reduced LV volumes and better LVEF. Recent reports have highlighted the impact of TEER on right ventricle performance even in patients without a significant improvement in LVEF, as in our cohort. In a study on 60 patients with secondary MR, pre-existing RV dysfunction (TAPSE < 16 mm and/or S’ RV < 10 cm/s) did not affect procedure efficacy outcome, and at 6 months, a significant change in RV function was reported [22]. A recent report on advanced echocardiographic evaluation of patients with mixed etiology MR showed early RV improvement immediately after the procedure and at 3 months of follow-up. The application of 3D echocardiography and speckle-tracking imaging let the authors identify reverse remodeling even in patients in which standard parameters suffered from low sensitivity [23]. 

In our study, we reported the comparison between primary and secondary MR cohort in terms of right sections remodeling. As expected, we found a more significant impact in RV and RA reverse remodeling, better longitudinal function, and lower PAPs in patients with functional MR and severe RV involvement. Interestingly, patients with primary MR and less compromised RV function also showed an impact from afterload reduction provided by TEER, reflected into reduced cavity dimensions and ameliorated contractile performance. Moreover, systematic application of advanced echocardiography evaluations such as RV deformational imaging could help in identifying subclinical RV dysfunction, which can be reverted by TEER procedure.

### 4.2. Left Ventricular and Atrial Chambers Remodeling

In our analysis, we reported LV remodeling in terms of LV end-diastolic linear internal diameter and end-diastolic and end-systolic volumes reduction without significant improvement of indices of systolic function. This is apparently contradictory, but some physiological considerations are worth mentioning. First, concomitant reduction in both end-diastolic and end-systolic parameters could lead to minor changes in LVEF. Second, it should be kept in mind that pre-procedural LVEF could be overestimated due to regurgitant volume. Third, in a progressive disease such as MR, the development of LV dysfunction is a continuous process that could be ongoing before the reduction of LVEF. From this perspective, the lack of further worsening in LVEF could be considered as a therapeutic success. On the contrary, LV diameters and volumes are strongly dependent on loading conditions, and therefore the reduction of LV preload after correction of MR induces a reduction of these parameters regardless of the initial mechanism. The geometrical changes in LV after TEER have been reported since the first experiences with TEER. In 2012, Scandura et al. reported a significant change in terms of sphericity index and LVEF after six months in a cohort of patients with mixed MR etiologies [24]. Similar results in terms of morpho-functional improvement of LV parameters were also reported in other observational studies on functional MR with severely reduced LVEF treated with TEER at 6 and 12 months of follow-up [21,25]. 

In our study, we report significant changes in geometrical reverse remodeling in the total population and in the primary MR subgroup. On the other hand, the small impact on the functional subgroup could be due to the less-dilated mean volumes and less-reduced mean LVEF in our FMR subgroup with respect to the population analyzed in previous studies [21,25]. Preprocedural and postprocedural predictors of LV remodeling after TEER have been extensively investigated in recent years. Residual MR, male sex, and baseline LVEF <20% proved to be the most relevant predictors of reverse remodeling with prognostic influence on cardiovascular outcomes [7,8]. Since almost all the patients in our research at 6 months had grade 1 residual MR, the relative influence of this specific factor has not been examined. Moreover, male sex percentage and median EF were differently represented in the primary and secondary cohorts, so we decided to perform the subgroup analysis only according to MR etiology.

As regards LA assessment, in the overall cohort, we report a significant reduction of LA volume and a trend to an improved reservoir function even without statistical significance. Chronic volumetric overload caused by regurgitating volume in the LA affects both volumetric and functional parameters. Restoration of the adequate competence of the mitral valve inducing LV unloading promotes mitral annulus size reduction and favors LV and LA reverse remodeling. The positive impact of TEER on LA remodeling is less consistent when considering the functional cohort in our population. In other reports on TEER in patients with FMR, a significant reduction of LA volume was achieved only after 12 months of follow-up [25]. Furthermore, a recent observational study by Toprak et al. reported that 3D echocardiographic evaluation of LA is more sensitive than 2D parameters in assessing reverse LA remodeling after TEER at 12 months [26]. In the same study, an improvement in LA function assessed by atrial speckle-tracking parameters was reported with a significant prognostic influence on morpho-functional impairment at baseline. We observed a more favorable atrial reverse remodeling associated with TEER in patients without history of AF as compared to patients with history of AF. It can be speculated that the lack of significant improvement in LA function in the AF cohort may be due to a higher degree of atrial cardiomyopathy, which prevented reverse remodeling. Indeed, AF promotes and sustains a vicious circle of atrial electrical and structural remodeling, possibly leading to “domestication” of the arrythmia and derangement in atrial volumes, geometry, function, and cellular structure [27,28,29]. Furthermore, the small sample size and the high number of patients in AF at baseline compared to previous studies could explain the limited extent of reverse remodeling [26].

### 4.3. Overall Hemodynamic Impact of TEER

Our study depicts the hemodynamic impact of TEER on patients with mixed MR etiologies. The most relevant change is the improvement of right heart hemodynamics (e.g., RV and RA dimensions, RV function, and PAPs) early after reduction of afterload with MR correction. The improvement of the afterload-dependent parameters occurs even without a significant change in terms of left ventricular function. The impact of the procedure on LV preload led to the improvement of preload-dependent parameters such as volumetric and linear measurements and LA volumes even without improvement in functional parameters.

### 4.4. Study Limitations

The main limitation of the study was the relatively small number of patients in the study population. Moving from this point, our findings are to be considered as hypothesis-generating, and additional research is necessary.

The observational and retrospective nature of the study has intrinsic limitations, linked to the selection bias and the possible presence of confounding factors. For some patients, it was not possible to recover and process the pre- and/or post-procedural echocardiographic parameters due to inadequate acoustic windows.

Additionally, as a result of the data collection process, we were only able to report on the usage of pharmacological classes before and after the procedure and were unable to assess dosage changes, which is a drawback, particularly for heart failure drugs.

## 5. Conclusions

TEER induces reverse remodeling involving both left and right chambers at mid-term follow-up. The most relevant impact is right chamber reverse remodeling that occurs even in the absence of the improvement of LV functional parameters. The MR mechanism and the history of AF significantly impact this process and should be considered for personalized patient management.

## Figures and Tables

**Figure 1 jpm-12-01916-f001:**
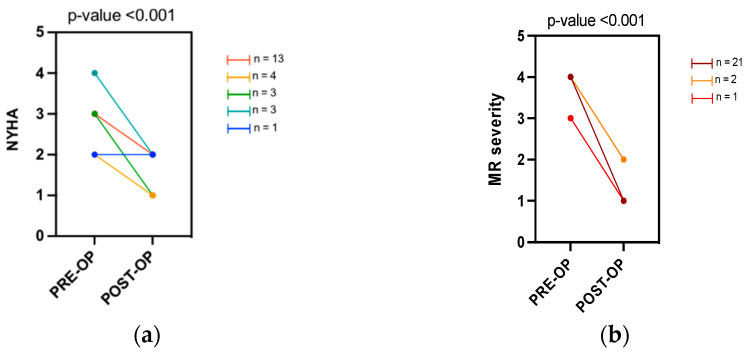
Change in NYHA class (**a**) and in mitral regurgitation severity (**b**) pre- and post-TEER. Legend: NYHA, New York Heart Association classification; pre-op, baseline evaluation; post-op, follow-up evaluation; MR mitral regurgitation.

**Figure 2 jpm-12-01916-f002:**
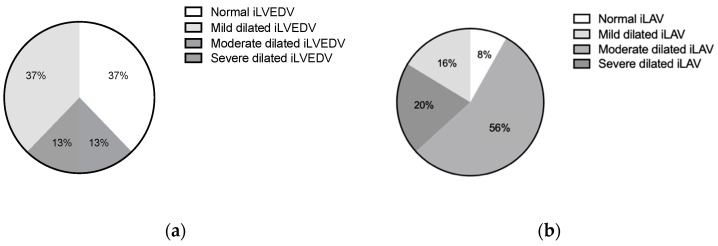
Grading of severity of left ventricular (**a**) and left atrial dilatation (**b**). Legend: iLVEDV, indexed left ventricular end-diastolic volume; iLAV, indexed left atrial volume.

**Figure 3 jpm-12-01916-f003:**
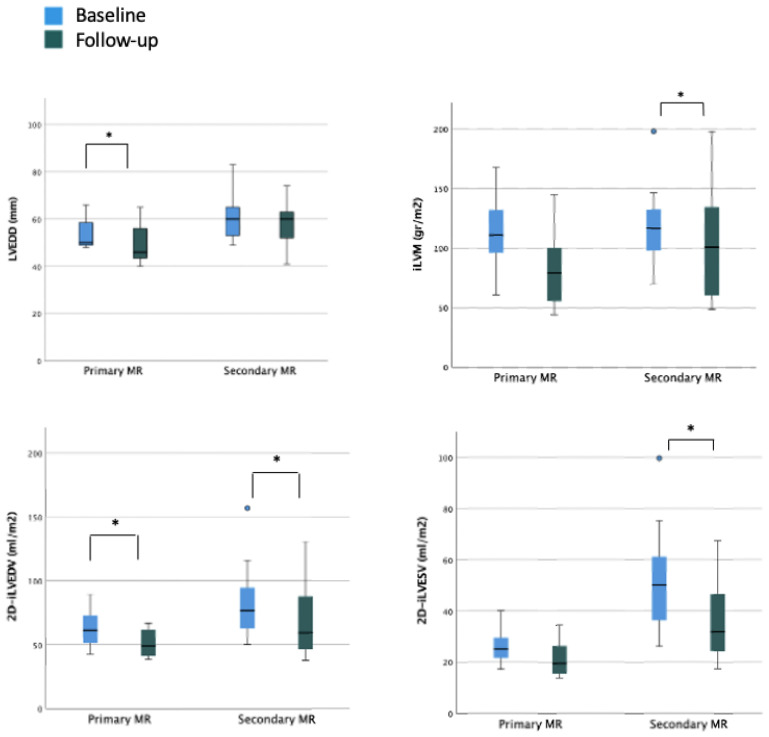
Ventricular and atrial remodeling after percutaneous mitral valve repair according to MR etiology. Legend: EPSPAP, estimated peak systolic pulmonary artery pressure; FAC, fractional area change; iLAV, indexed left atrial volume; iLVM, indexed left ventricular mass; iLVEDV, indexed left ventricular end-diastolic volume; iLVESV, indexed left ventricular end-systolic volume; LVEDD, left ventricular end-diastolic diameter; RA, right atrium; RVD, right ventricular diameter; RV GLS FW, right ventricle free wall longitudinal strain; TAPSE, tricuspid annular plane systolic excursion; TR, tricuspidal regurgitation. * = *p* < 0.05.

**Figure 4 jpm-12-01916-f004:**
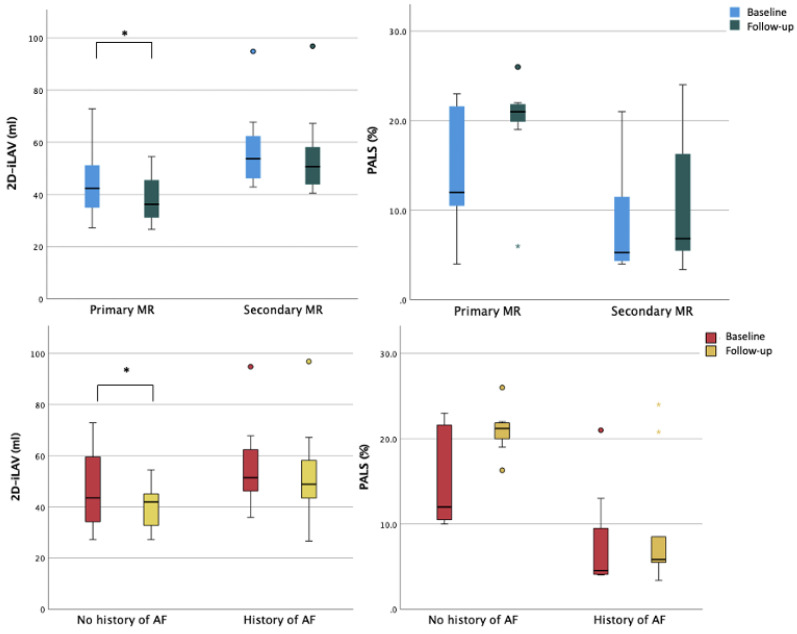
Left atrial remodeling according to mitral regurgitation etiology and history of atrial fibrillation, respectively. Legend: AF, atrial fibrillation; iLAV, indexed left atrial volume; MR, mitral regurgitation; PALS, peak left atrial longitudinal strain. * = *p* < 0.05.

**Table 1 jpm-12-01916-t001:** Baseline clinical assessment.

Clinical Characteristics	Total Population (*n* = 24)	Primary MR (*n* = 11)	Secondary MR (*n* = 13)	*p*-Value
Males, n (%)	15 (62.5)	3 (27)	12 (92)	0.001
Age (years), mean (± SD)	78.5 (±7.6)	77.09 (±7.8)	79.7 (±7.5)	0.404
BSA Mosteller (m^2^), mean (± SD)	1.8 (±0.2)	1.7 (±0.2)	2.0 (±0.2)	0.019
BMI (Kg/m^2^), mean (± SD)	26.4 (± 4.1)	26.2 (±4.3)	26.7 (±4.2)	0.398
EuroSCORE II (%), median [IQR]	4 [3.1–6.2]	3.7 [2.8–4.5]	4.8 [3.5–9.1]	0.104
STS score (%), median [IQR]	3.5 [1.7–4.8]	2.5 [1.2–5.8]	3.6 [2.6–4.7]	0.384
NYHA class III, n (%)	17 (70.8)	7 (63)	10 (77)	0.476
NYHA class IV, n (%)	7 (29.2)	4 (37)	3 (23)	0.476
Dyslipidemia, n (%)	13 (54.1)	6 (54)	7 (54)	0.973
Diabetes, n (%)	7 (29.1)	2 (19)	5 (38)	0.276
Hypertension, n (%)	16 (66.6)	7 (64)	9 (69)	0.772
Smoking history, n (%)	7 (29.1)	2 (19)	5 (38)	0.276
Previous CABG, n (%)	4 (16.6)	1 (9)	3 (23)	0.360
Previous PCI, n (%)	13 (54.1)	4 (36)	9 (69)	0.107
Previous AMI, n (%)	6 (25)	1 (9)	5 (38)	0.098
PM, n (%)	2 (8.3)	0 (0)	2 (15)	0.174
Previous AV surgery, n (%)	1 (4.1)	0 (0)	1 (7)	0.347
Previous TAVI, n (%)	1 (4.1)	0 (0)	1 (7)	0.347
History of AF, n (%)	14 (58.3)	5 (45)	9 (69)	0.239
DHP-CCBs, n (%)	9 (37,5)	4 (36)	5 (38)	0.916
ACE-I, n (%)	4 (16.6)	2 (19)	2 (15)	0.855
ARB, n (%)	8 (33,3)	2 (19)	6 (46)	0.148
MRAs, n (%)	22 (91.6)	10 (90)	12 (92)	0.902
Diuretics, n (%)	23 (95.8)	11 (100)	12 (92)	0.347
Beta-blockers, n (%)	21 (87.5)	9 (82)	12 (92)	0.439

Legend: ACE-I, angiotensin-converting-enzyme inhibitors; AF, atrial fibrillation; AMI, acute myocardial injury; ARBs, angiotensin II receptor blockers; AV, aortic valve; BSA, body surface area; BMI, body mass index; CABG, coronary artery bypass grafting; DHP-CCBs, dihydropyridine–calcium channel blockers; MRAs, mineralocorticoid receptor antagonists; MV, mitral valve; NYHA, New York Heart Association classification; PCI, percutaneous coronary intervention; PM, pacemakers; STS score, Society of Thoracic Surgeons score; TAVI, transcatheter aortic valve implantation; TIA, transient ischemic attack.

**Table 2 jpm-12-01916-t002:** Baseline echocardiographic characteristics.

	Total Population (*n* = 24)	Primary MR (*n* = 11)	Secondary MR (*n* = 13)	*p*-Value
LVEDD (mm)	57.5 [50.0–64.5]	50.0 [49.0–59.0]	60.0 [53.0–65.0]	0.059
LVESD (mm)	42.5 [37.8–50.5]	42.0 [37.0–49.0]	45.0 [40.5–52.0]	0.450
RWT	0.32 [0.27–0.37]	0.32 [0.29–0.38]	0.31 [0.26–0.37]	0.424
iLVM (gr/m^2^)	116.6 [95.7–132.4]	111.1 [95.1–142.9]	116.9 [96.2–136.4]	0.820
2D-iLVEDV (mL/m^2^)	67.5 [57.7–86.8]	61.4 [45.4–77.7]	76.4 [60.5–94.9]	0.082
2D-iLVESV (mL/m^2^)	35.4 [25.4–51.0]	25.2 [21.2–39.8]	50.2 [35.5–67.1]	0.001
2D-LVEF (%)	48.8 [39.8–58.1]	59.1 [50.7–61.5]	42.4 [35.9–48.2]	<0.001
LV-S’ (TDI) (cm/s)	8.0 [6.0–8.5]	8.0 [7.5–9.5]	6.5 [5.3–8.0]	0.127
2D-iLAV (mL/m^2^)	50.5 [42.5–62.3]	42.3 [34.2–51.7]	53.8 [45.2–62.6]	0.052
PALS (%)	11.0 [4.5–19.0]	12.0 [4.0–17.0]	10.0 [5.0–17.0]	0.786
TR Vmax (m/s)	2.9 [2.5–3.1]	3.0 [2.5–3.1]	2.9 [2.5–3.2]	0.786
EPSPAP (mmHg)	40.0 [35.0–50.0]	40.0 [35.0–50.0]	42.5 [31.3–53.8]	0.928
Basal RVD (mm)	38.5 [35.0–40.8]	35.0 [31.0–40.0]	39.0 [37.0–43.0]	0.106
Mid-cavity RVD (mm)	29.5 [25.2–34.0]	27.0 [25.0–34.0]	30.0 [26.0–35.0]	0.459
Longitudinal RVD (mm)	64.0 [55.2–69.8]	59.0 [52–68]	65.0 [60.5–71.0]	0.119
RA Area (cm^2^)	19.5 [15.3–23.0]	18.0 [15.0–23.0]	21.0 [17.5–25.0]	0.167
FAC (%)	35.5 [30.0–39.8]	37.0 [35.0–44.0]	34.0 [30.0–37.0]	0.051
TAPSE (mm)	14.0 [13.0–19.2]	20.0 [17.0–24.0]	14.0 [13.0–14.0]	0.003
RV S’ (TDI) (cm/s)	9.0 [7.0–12.0]	9.0 [8.0–13.7]	7.0 [6.0–10.8]	0.127
RV-FWLS (%)	−19.0 [−11.0–−22.0]	−21.0 [−15.1–−23.0]	−15.0 [−9.9–−19.8]	0.127

Legend: EPSPAP, estimated peak systolic pulmonary artery pressure; FAC, fractional area change; iLAV, indexed left atrial volume; iLVM, indexed left ventricular mass; iLVEDV, indexed left ventricular end-diastolic volume; iLVESV, indexed left ventricular end-systolic volume; iRVEDV, indexed right ventricular end-diastolic volume; iRVESV, indexed right ventricular end-systolic volume; LAEF, left atrial ejection fraction; LVEDD, left ventricular end-diastolic diameter; LVESD, left ventricular end-systolic diameter; LVEF, left ventricular ejection fraction; LV S’, systolic velocity at the mitral annular; PALS, peak left atrial longitudinal strain; RA, right atrium; RVD, right ventricular diameter; RV-FWLS, right ventricle free wall longitudinal strain; RWT, relative wall thickness; TAPSE, tricuspid annular plane systolic excursion; TR, tricuspidal regurgitation.

**Table 3 jpm-12-01916-t003:** Baseline and follow-up assessment.

	Baseline Assessment, Median [IQR]	Follow-Up Assessment, Median [IQR]	*p*-Value
LVEDD (mm)	57.5 [50.0–64.5]	55.0 [45.3–60.0]	<0.001
LVESD (mm)	42.5 [37.8–50.5]	44.5 [38.3–48.0]	0.660
RWT	0.32 [0.27–0.37]	0.34 [0.26–0.38]	0.201
iLVM (gr/m^2^)	116.6 [95.7–132.4]	90.2 [56.7–113.0]	0.006
2D-iLVEDV (mL/m^2^)	67.5 [57.7–86.8]	53.5 [42.1–65.8]	<0.001
2D-iLVESV (mL/m^2^)	35.4 [25.4–51.0]	27.1 [19.7–42.9]	0.002
2D-LVEF (%)	48.8 [39.8–58.1]	49.6 [43.8–60.6]	0.248
LV-S’ (TDI) (cm/s)	8.0 [6.0–8.5]	8.0 [7.5–9.0]	0.053
2D-iLAV (ml/m^2^)	50.5 [42.5–62.3]	44.1 [37.4–56.4]	0.004
PALS (%) [N = 8]	11.0 [4.5–19.0]	14.6 [4.6–23.5]	0.123
TR Vmax (m/s)	2.9 [2.5–3.1]	2.2 [1.8–2.6]	<0.001
EPSPAP (mmHg)	40.0 [35.0–50.0]	20.0 [20.0–30.0]	<0.001
Basal RVD (mm)	38.5 [35.0–40.8]	33.5 [30.5–37.0]	<0.001
Mid-cavity RVD (mm)	29.5 [25.2–34.0]	27.0 [22.3–29.8]	0.001
Longitudinal RVD (mm)	64.0 [55.2–69.8]	58.5 [53.0–61.0]	<0.001
RA Area (cm^2^)	19.5 [15.3–23.0]	16.0 [13.2–20.0]	0.002
FAC (%)	35.5 [30.0–39.8]	44.0 [40.0–46.5]	<0.001
TAPSE (mm)	14.0 [13.0–19.2]	18.5 [16.0–21.0]	<0.001
RV S’ (TDI) (cm/s)	9.0 [7.0–12.0]	9.0 [9.0–12.0]	0.049
RV-FWLS (%)	−19.0 [−11.0–−22.0]	−21.0 [−16.0–−23.5]	0.006

Legend: EPSPAP, estimated peak systolic pulmonary artery pressure; FAC, fractional area change; iLAV, indexed left atrial volume; iLVM, indexed left ventricular mass; iLVEDV, indexed left ventricular end-diastolic volume; iLVESV, indexed left ventricular end-systolic volume; iRVEDV, indexed right ventricular end-diastolic volume; iRVESV, indexed right ventricular end-systolic volume; LAEF, left atrial ejection fraction; LVEDD, left ventricular end-diastolic diameter; LVESD, left ventricular end-systolic diameter; LVEF, left ventricular ejection fraction; LV S’, systolic velocity at the mitral annular; PALS, peak left atrial longitudinal strain; RA, right atrium; RVD, right ventricular diameter; RV-FWLS, right ventricle free wall longitudinal strain; RWT, relative wall thickness; TAPSE, tricuspid annular plane systolic excursion; TR, tricuspidal regurgitation.

## Data Availability

Data available on request due to restrictions (e.g., privacy or ethical).

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
