# Peer review of "Ventricular and Atrial Remodeling after Transcatheter Edge-to-Edge Repair: A Pilot Study"

_jpm, 2022, doi:10.3390/jpm12111916_

Round 1

Reviewer 1 Report

The authors have presented a useful studly of the effects of the MitraClip procedure in a high surgical risk cohort.  Their cohort is small in part due to the limitations of echocardiographic imaging in this elderly population. Suggestions for improvement in the manuscript are few.

It should be stated if the patients were in sinus rhythm for both imaging studies.

For those that had been in atrial fibrillation it would be of interest to better characterize this sub-group; durations of AF events, whether recent or remote, and therapy to maintain NSR.

Was the grading of the severity of MR blinded?

Since symptoms improved, were there changes in the medical therapy following the MitraClip procedure?

Author Response

We thank the reviewers for the useful comments and suggestions. All the changes are reported in the “track changes” version of the manuscript and also reported below.

Reviewer: The authors have presented a useful studly of the effects of the MitraClip procedure in a high surgical risk cohort.  Their cohort is small in part due to the limitations of echocardiographic imaging in this elderly population. Suggestions for improvement in the manuscript are few.

It should be stated if the patients were in sinus rhythm for both imaging studies.

Authors: We thank you for your suggestion. All the patients with AF history were in AF during baseline and follow-up evaluation. We added this phrase in the text

“Patients with history of AF had persistent-long term or permanent AF, so baseline and follow-up echocardiographic assessment were performed during AF”.

Reviewer: For those that had been in atrial fibrillation it would be of interest to better characterize this sub-group; durations of AF events, whether recent or remote, and therapy to maintain NSR.

Authors: Unfortunately, data about previous AF events or previous therapy for rhythm control were not collected in the study protocol.

Reviewer: Was the grading of the severity of MR blinded?

The echocardiographic evaluations were performed by the same operator before and after the procedure. We did not opt for an external evaluation because our aim was to provide a real-life clinical experience. We reported in the methods section the modality of mitral regurgitation grading adopted in the study.

“Due to the significant proportion of eccentric MR in the population, a semi-quantitative method with color Doppler was employed for the baseline evaluation and the post-implant grading.”

Reviewer: Since symptoms improved, were there changes in the medical therapy following the MitraClip procedure?

Authors: Thank you for your comment, this specific aspect is surely worth mentioning. No significant changes in the medical therapy were reported before and after the procedure. We reported it in the text in the supplementary table n°1. Due to the data collecting procedure, we reported pharmacological class use and cannot evaluate changes in medications dosages.

“No significant changes in the medical therapy were reported before and after the procedure. (supplementary table 1)”

Reviewer 2 Report

This analysis by Albini and colleagues presents echocardiographic follow-up in 24 patients from a single institution after MitraClip focusing on LV, RV, and LA chamber size changes.

Overall, the manuscript is well-written and clear to follow.  The analyses are sound and well-presented.  Tables 3 and 4 are little too big with too much information to follow clearly.  Some of this information might be more clearly displayed in graphical form.  Certainly, when testing that many variables you are bound to find some that are statistically significant.  The sample size is too small to adequately make that many comparisons of that many different variables.  

This is an important topic and we need a better understanding of echo-based changes that occur in the heart after transcatheter mitral valve repair.  However, given the small size of the sample in this paper, these results can only be hypothesis generating.  The manuscript should acknowledge this.  

This sentence in the abstract is a little confusing and should be reworded: "LVEDD and LVEDV improved in primary 19 MR cohort whereas in secondary MR a significant reduction in LVEDV and LVESV with a nonsig-20 nificant functional improvement were found."

I think there may be an error in this sentence: "STI analyses was performed using commercially 82 available AutoStrain software Philips Healthcare, for LA and RV-strain software Philips 83 Healthcare for RV."  It should be reworded.

In the US, we are using transcatheter edge-to-edge repair (TEER) as the accepted term for MitraClip. The authors should consider switching PMVR to TEER throughout.

I think the most important findings of this manuscript are the changes to RV function and size this soon after MitraClip.  This needs to be highlighted as the main finding of the paper as I feel this is the most clinically significant finding from this study.  

Author Response

We thank the reviewers for the useful comments and suggestions. All the changes are reported in the “track changes” version of the manuscript and also reported below.

Reviewer: This analysis by Albini and colleagues presents echocardiographic follow-up in 24 patients from a single institution after MitraClip focusing on LV, RV, and LA chamber size changes.

Overall, the manuscript is well-written and clear to follow.  The analyses are sound and well-presented.  Tables 3 and 4 are little too big with too much information to follow clearly.  Some of this information might be more clearly displayed in graphical form.  Certainly, when testing that many variables you are bound to find some that are statistically significant.  The sample size is too small to adequately make that many comparisons of that many different variables.  

Authors: Thanks for your suggestion. We reported data from table 4 in graphical form for a clearer interpretation of baseline/follow-up comparison according to MR etiology. For space optimization, we reported only significant comparisons. All the comparisons are reported in the table in the supplementary material (table S2, former table 4).

Reviewer: This is an important topic and we need a better understanding of echo-based changes that occur in the heart after transcatheter mitral valve repair.  However, given the small size of the sample in this paper, these results can only be hypothesis generating.  The manuscript should acknowledge this.  

Authors: We appreciate the reviewer's thoughtful advice. Although we noted that our study's primary restriction was its limited numerosity in the section on limitations, we went on to emphasize more this aspect.

“The main limitation of the study was the relatively small number of patients in the study population. Moving from this point, our findings are to be considered as hypothesis-generating and additional research is necessary”

Reviewer: This sentence in the abstract is a little confusing and should be reworded: "LVEDD and LVEDV improved in primary 19 MR cohort whereas in secondary MR a significant reduction in LVEDV and LVESV with a nonsig-20 nificant functional improvement were found."

Authors: We apologize for the mistype. We amended it.

LVEDD and LVEDV improved in primary MR cohort whereas in secondary MR a significant reduction in LVEDV and LVESV was found without a significant functional improvement

Reviewer:I think there may be an error in this sentence: "STI analyses was performed using commercially 82 available AutoStrain software Philips Healthcare, for LA and RV-strain software Philips 83 Healthcare for RV."  It should be reworded.

 Authors: We apologize for the misleading formulation. We rephrased the sentence as follows:

STI analyses was performed using commercially available AutoStrain LA and AutoStrain RV softwares (Philips Healthcare).

Reviewer: In the US, we are using transcatheter edge-to-edge repair (TEER) as the accepted term for MitraClip. The authors should consider switching PMVR to TEER throughout.

Authors: We thank the reviewer for the suggestion. We changed the terminology throughout the text.

Reviewer: I think the most important findings of this manuscript are the changes to RV function and size this soon after MitraClip.  This needs to be highlighted as the main finding of the paper as I feel this is the most clinically significant finding from this study.  

Authors: We thank the reviewer for his/her suggestion. We rearranged the discussion in order to emphasize the most relevant impact of TEER on right sections.

Reviewer 3 Report

The manuscript „Ventricular and atrial remodeling after percutaneous mitral valve repair: a pilot study” by Albini et al. describes remodeling of the left and right atria and ventricles after mitral transcatheter edge-to-edge-repair in 24 patients assessed by speckle-tracking. The strengths of the manuscript are the application of speckle-tracking for assessment and the evaluation of left and right heart parameters.

However, there are some points that should merit consideration:

1)    As the authors point out themselves, 24 patients is a very low number. Especially when taking into account that these numbers were acquired over 4 years (2017-2021). Of the initial 35 patients, 11 patients had to be excluded, so the 24 patients are not consecutive, as mentioned on p.9, line 249.

2)    The authors state, that “Nowadays, few 46 data on cardiac chambers reverse remodeling and function after PMVR are available. 47 [5,6]“. But much more publications on this topic are available, such as 12, 13, 14, 26, 27, and additional publications that have not been mentioned such as Nita et al. Predictors of left ventricular reverse remodeling after percutaneous therapy for mitral regurgitation with the MitraClip system, Catheter Cardiovasc Interv. 2020;96:687–697. or Grayburn PA, Foster E, Sangli C, et al. Relationship between the mag- nitude of reduction in mitral regurgitation severity and left ventricular and left atrial reverse remodeling after MitraClip therapy. Circulation. 2013;128(15):1667-1674. 

3)    The impact of residual or even recurrent MR after TEER has not be analysed

4)    The impact of changes of the heart-failure medication has not been analysed.

Minor comment: On page 3, Table 1: NYHA instead of NHYA.

All in all, the authors have to concentrate more the postprocedural influential factors that might impact on their results and existing literature should be discussed more extensively before this manuscript should be considered for publication. But due to the fact that speckle-tracking was used for analysis, this manuscript features some novelty, which is a strength.

Author Response

We thank the reviewers for the useful comments and suggestions. All the changes are reported in the “track changes” version of the manuscript and also reported below.

The manuscript „Ventricular and atrial remodeling after percutaneous mitral valve repair: a pilot study” by Albini et al. describes remodeling of the left and right atria and ventricles after mitral transcatheter edge-to-edge-repair in 24 patients assessed by speckle-tracking. The strengths of the manuscript are the application of speckle-tracking for assessment and the evaluation of left and right heart parameters.

However, there are some points that should merit consideration:

Reviewer: 1)    As the authors point out themselves, 24 patients is a very low number. Especially when taking into account that these numbers were acquired over 4 years (2017-2021). Of the initial 35 patients, 11 patients had to be excluded, so the 24 patients are not consecutive, as mentioned on p.9, line 249.

Authors: We thank the reviewer for the thoughtful comment. We amended it in the text.

Reviewer: 2)    The authors state, that “Nowadays, few 46 data on cardiac chambers reverse remodeling and function after PMVR are available. 47 [5,6]“. But much more publications on this topic are available, such as 12, 13, 14, 26, 27, and additional publications that have not been mentioned such as Nita et al. Predictors of left ventricular reverse remodeling after percutaneous therapy for mitral regurgitation with the MitraClip system, Catheter Cardiovasc Interv. 2020;96:687–697. or Grayburn PA, Foster E, Sangli C, et al. Relationship between the mag- nitude of reduction in mitral regurgitation severity and left ventricular and left atrial reverse remodeling after MitraClip therapy. Circulation. 2013;128(15):1667-1674. 

Authors: We thank the reviewer for the observation. We rephrased the sentence as follows:

“Although some data on the reverse remodeling and function of heart chambers following TEER are available, [5,6] we think that further research and reliable data should be collected to reach a more detailed comprehension of left and right sections interplay in the process.”

Reviewer 3)    The impact of residual or even recurrent MR after TEER has not be analysed

Authors: Thank you for asking for this aspect that is worth discussing. Since 22 patients on the total reached a grade 1 mitral regurgitation severity at 6 months, we didn’t consider relevant to analyse the impact of residual MR on the remodeling process.

Reviewer: 4)    The impact of changes of the heart-failure medication has not been analysed.

Authors: Thank you for your comment, this specific aspect is surely worth mentioning. We reported baseline therapy in order to emphasize that patients were treated with guideline-directed medical therapy. Especially for patients with reduced EF and secondary MR, this is strictly recommended by current guidelines before considering patients for TEER. Moreover, no significant changes in the medical therapy were reported before and after the procedure. We reported it in the text and in the supplementary table n°1. Due to the data collecting procedure, we reported pharmacological class use and cannot evaluate changes in medications dosages.

Reviewer: Minor comment: On page 3, Table 1: NYHA instead of NHYA.

Authors: We amended it.

Reviewer: All in all, the authors have to concentrate more the postprocedural influential factors that might impact on their results and existing literature should be discussed more extensively before this manuscript should be considered for publication. But due to the fact that speckle-tracking was used for analysis, this manuscript features some novelty, which is a strength.

Authors: Preprocedural and postprocedural predictors of LV remodeling after TEER have been extensively investigated in recent years. Residual MR, male sex and baseline LVEF <20% resulted to be the most relevant predictors of reverse remodeling with prognostic influence on cardiovascular outcomes. (DOI: 10.1002/ccd.28779, DOI: 10.1161/CIRCULATIONAHA.112.001039). Since in our analysis at 6 months nearly all the patients showed grade 1 residual MR at 6 months, the relative impact of this particular factors has not been analyzed. Moreover, male sex percentage and median EF were differently represented in primary and secondary cohort, so we decided to perform the subgroup analysis only according to MR etiology.

Round 2

Reviewer 3 Report

The manuscript has relevantly improved during the review process. 

But the authors still do not cite all relevant publications on the topic such as Grayburn et al, Nita et al. etc.!

Regarding HF medication, changes were not analyzed. This has to be mentioned as a relevant limitation, since the hospitalization for TEER is often a chance to intensify HF medication, which impacts on reverse remodeling.

Author Response

We thank the reviewer for the useful comments and suggestions. All the changes are reported in the “track changes” version of the manuscript and also reported below.

Reviewer: The manuscript has relevantly improved during the review process. 

But the authors still do not cite all relevant publications on the topic such as Grayburn et al, Nita et al. etc.!

Authors: We appreciate the suggestion, and we have included the suggested publications in the introduction and discussion sections as references.

“Preprocedural and postprocedural predictors of LV remodeling after TEER have been extensively investigated in recent years. Residual MR, male sex and baseline LVEF <20% resulted to be the most relevant predictors of reverse remodeling with prognostic influence on cardiovascular outcomes. [7,8] Since almost all the patients in our analysis at 6 months had grade 1 residual MR, the relative influence of this specific factor has not been examined. Moreover, male sex percentage and median EF were differently represented in primary and secondary cohort, so we decided to perform the subgroup analysis only according to MR etiology.”

Reviewer: Regarding HF medication, changes were not analyzed. This has to be mentioned as a relevant limitation, since the hospitalization for TEER is often a chance to intensify HF medication, which impacts on reverse remodeling.

Authors: Thanks for your observation. We integrated the limitations section as follows:

“Additionally, as a result of the data collection process, we were only able to report on the usage of pharmacological classes before and after the procedure and were unable to assess dosage changes, which is a drawback, particularly for HF drugs.”